# Improving Concentration and Academic Performance of a Mathematically Talented Student with ASD/ADHD: An Enrichment Program

Kun-Ming Lien, Ching-Chih Kuo and Hung-Lun Pan *

Department of Special Education, National Taiwan Normal University, Taipei 10610, Taiwan; kevin1218@apps.ntpc.edu.tw (K.-M.L.); kaykuo@ntnu.edu.tw (C.-C.K.)
* Correspondence: haroldpan@apps.ntpc.edu.tw

**Abstract:** This study explored whether computer-assisted, project-based learning instruction can help a twice-exceptional (2e) student increase classroom concentration, mathematical concepts, and problem-solving skills. This research used a case study design. The researchers analyzed data collected from student and teacher interviews, behavioral records, and task performances. The result showed that the incidence of misbehaviors decreased from more than ten times to fewer than three times per hour. According to the Flow Short Scale (FSS), reports of his peers, and the case management teacher, the participant with autism spectrum disorder (ASD) was highly attentive during the project. As for his academic performances, the rubric and scoring results from the instructors suggested that this participant performed very well in data representation, logical thinking, and mathematical thinking. However, he obtained a low score in flow control because of a lack of experience. His peers noted that he understood the mathematics concept of the tasks, was highly proficient in Scratch, completed a considerable portion of his work, and was willing to share the details of his works thoroughly. His peers in the project praised his learning attitude and the quality of his work highly.

**Keywords:** twice exceptionality (2e); Autism Spectrum Disorder (ASD); mathematics problem solving; project-based learning (PBL)

## 1. Introduction

### 1.1. Importance of Learning Opportunity and Educational Equity

Educational equity is a persistent societal concern [1–4]. According to Gonzalez (2001), "Education is the great equalizer in a democratic society, and if people are not given access to a quality education, then what we are doing is creating an underclass of people". When all students have equal access to high-quality educational resources and opportunities, they are able to realize their full potential and contribute to the economic and social development of society [5]. Lack of access to adequate educational resources and opportunities because of factors such as economics, race, gender, cultural background, or disabilities leads to social unfairness and prevents students from realizing their full potential. Since education is a fundamental human right, ensuring educational opportunity and equity is teacher's duty [6,7].

Educational equity is a major focus of the United Nations Educational, Scientific and Cultural Organization "Education for All Global Monitoring Report". The Organization for Economic Cooperation and Development also believes that children are the foundation of the future society. Increasing fairness entails providing learners with capabilities and flexibility and allowing them to pursue their education in accordance with changing environments, interests, and needs. Confucianism has influenced the educational values of East Asia. Confucian educational principles from 2500 years ago emphasized "teaching without discrimination" and "teaching according to aptitude".

Educational equity is also emphasized in Article 7 of Taiwan's 2013 Educational Fundamental Act:

"All people, regardless of their sex or gender, ages, abilities, geographic locality, ethnic group(s), religious beliefs or political ideas, social or economic status or other conditions, have equal opportunity for receiving education. Special protection on the education for indigenous peoples, the physically or mentally challenged or other disadvantaged groups shall be provided with considerations of their autonomy and special characteristics in accordance with relevant laws and regulations to support their development".

### 1.2. Education Equity for Twice-Exceptional Students

According to Rawls (1971), "social and economic inequalities of wealth and authority are only just if they result in compensating benefits for everyone, particularly the least advantaged in society" [8]. The term "twice-exceptional" (2E) was coined by Whitmore to describe students who have extraordinary talents or cognitive abilities but are limited in their ability to develop them because of impairments.

Students with twice exceptionality are those who have coexisting giftedness and disabilities in one or more domains that need support from both gifted and disability education [9–12]. These students may also experience discrimination or prejudice because of attributes related with their physical and mental differences [13]. Individuals may be disadvantaged in numerous respects and thus may be affected by multiple equity gaps [4].

In Taiwan, for the identification of students with special education needs, local authorities should set up a Special Education Students Diagnosis and Placement Counseling Committee (briefly called DPCC), inviting scholars and experts, educational and school administrators, delegates of teacher organizations, parents, professionals of special education, and delegates of related institutions and groups to participate in diagnosis, placement, replacement, and counseling.

According to Chen et al [14,15] in Taiwan, the statistics acquired from the Ministry of Education's Special Education Transmit Net reported that there were 376 students identified as twice exceptional at the 1–12 grade levels in 2019. In a sample of 100 gifted and talented students, 1.34% of the population was identified as twice exceptional; in a sample of 100 students with disability, 0.4% of the population was identified as twice exceptional; in a sample of 100 school-age students, only 0.015% of the population was identified as twice exceptional [14,15]. This number (0.015%) is considerably lower than the estimated 6% prevalence in the United States [16]. Without suitable support and services, these students may struggle to learn and participate in school activities, thereby leading to educational inequality [16].

In addition to the United States and Taiwan, it is worth noting that other countries, such as Spain and Ireland in Europe, also address the needs of dual exceptional (gifted and special needs) students in their legislation. Spain explicitly includes provisions for the care of gifted students and their educational needs in its laws. For instance, the "Royal Decree 696/1995" (BOE, 2 June) regulates the conditions for educational attention to students with temporary or permanent special needs associated with educational history, including those arising from giftedness, mental disability, or motor or sensory impairments. Meanwhile, although Ireland's laws do not specifically mention gifted students, they emphasize the provision of education and support services that are aligned with the individual needs and abilities of all students, including those with special educational needs [17].

Therefore, ensuring that twice-exceptional students receive appropriate support and services to aid in their learning and participation in school activities is integral to educational equity.

### 1.3. Excellence Gap in Twice-Exceptional Students

Klingner (2022) noted that a gap may exist between twice-exceptional students and their peers [18]. Ziegler et al. (2021) [4] proposed greater transparency for equity gaps in education and their considerable variance within groups. Greater transparency is also

required for groups of twice-exceptional gifted students with disabilities. According to a National Education Association report in the United States [16], many seemingly average students are students whose gifts and disabilities mask one another. Discrepancies between their strengths and weaknesses in school may cause frustration, which can lead to social, emotional, and behavioral problems. Teaching adjustments for twice-exceptional students may include adjustments in assessment, teaching, and learning style, including the following [19]:

1.  In-depth explorations within interest areas;
2.  Adopting interdisciplinary themes;
3.  Providing real-world, problem-based learning experiences;
4.  Allowing students to self-select projects;
5.  Providing open-ended challenges;
6.  Allowing different pathways for learning;
7.  Implementing compensation strategies for areas of weakness to help students understand their strengths and weaknesses and enhance their active participation;
8.  Teaching study skills, such as decoding, note taking, and organization.

Appropriate teaching strategies enable twice-exceptional students to recognize their interests and superior abilities, nurture their potential, and enhance their ability to integrate their strengths and weaknesses, thereby enabling them to thrive academically.

### 1.4. The Comorbidity of Autism Spectrum Disorder and ADHD

Both ASD and ADHD are neurodevelopmental disorders. There are behavioral, biological, and neuropsychological overlaps between the two diseases; 50% to 70% of individuals with Autism Spectrum Disorder also have comorbid ADHD [20,21]. Prior to the Diagnostic and Statistical Manual for Mental Disorders—5th edition (DSM-5) in 2013, clinicians were unable to make an ADHD diagnosis in the context of ASD. It was presumed that any symptoms of inattention and/or hyperactivity–impulsivity were secondary to ASD and not due to an additional ADHD diagnosis [22].

The problems with ASD-ADHD comorbidity are wide-ranging and may affect areas such as social skills, language skills, attention, activity levels, and compulsive behaviors of an individual. Children with ASD + ADHD were generally rated to have more severe anxiety symptoms compared to the other groups [23]. While neither autism nor ADHD is curable, there are many strategies used to treat the symptoms. These range from behavioral treatment plans to psychopharmacological measures. Due to the observation that the symptoms of ADHD such as inattention, distractibility, and impulsiveness can be observed in children with autism, many studies have investigated the effectiveness of treatments for ADHD in children with autism [24]. Although most learning disabilities are lifelong conditions, using a strengths-based, talent-focused approach for twice-exceptional learners with a feeling of safety, value, and acceptance, many individuals not only gain coping skills but, as many adults have demonstrated, also can learn to thrive in spite of the difficulties [25,26]. After collecting the information, the researcher decided to use programming which is the interests and advantages of the individual to carry out the course design.

### 1.5. Teaching Programming to Students with Special Needs

After the field of artificial intelligence was established at Dartmouth in 1956, computer scientists recognized the connection between mathematics and computer science and promoted the use of information technology, programming, and computer-assisted learning at all levels of education [27,28]. The mathematics and science fields have progressed rapidly over time with the development of new tools. The Common Core State Standards Initiative guidelines stipulate that students should be able to use technological tools to explore and deepen their understanding of concepts, which illustrates how computational thinking is applied in mathematics and science classrooms [29]. Research on how programming applies to mathematical learning has explored areas including student motivation to learn

mathematics, student performance in mathematics, collaboration between students, and the changing role of teachers [30]. Computational tools enhance the learning of mathematical and scientific content. Mathematics and science also provide a meaningful context for computational thinking [29]. The Massachusetts Institute of Technology designed Scratch as a foundational programming language for children from economically disadvantaged backgrounds. Scratch was developed to promote children's computational thinking and critical thinking skills. Children can easily learn to use this tool because it has a user-friendly interface, it has been translated into more than 70 languages, and lifetime access to it is free [31].

Scratch is popular in programming education in primary and secondary schools [32]. The English primary school project ScratchMaths consists of a 2-year curriculum for students in the fifth and sixth grades. The content of this project is based on the British National Computer and Mathematics Primary School Curriculum. Carefully selected core concepts of computer programming and mathematics are taught to students in the aforementioned project to enable them to learn computational and mathematical thinking simultaneously in the Scratch language environment [33]. However, because of the inconvenience of Scratch for complex combinations of logical conditions and the lack of parallel control, many educators use Python for teaching. Python programs have concise instructions and easily comprehensible syntax. Beginners may complete almost all programming tasks in Python without any specific hardware requirements [34].

Numerous special education studies have used computers as a therapy or an educational resource to assist children with Autism Spectrum Disorder [35–37]. Studies have also noted that computational thinking and computer science are suitable teaching fields for students with disabilities. Computational thinking and computer science activities are convenient to implement and can benefit students with disabilities in various educational settings, particularly mathematics education [33,38–45]. Many special education studies have been conducted in the Scratch environment. For example, Munoz et al. (2016) [44] taught adolescents with ASD to use the Scratch language for game design through workshops to help ASD students overcome obstacles. Through the use of an immersive virtual network technique, students with ASD who face difficulties in socializing can avoid direct face-to-face interactions [45], which provides them with a more comfortable learning environment and improved autonomy over their learning pace.

### 1.6. Supporting the Programming Education of Twice-Exceptional Students through Project-Based Learning

Project-based learning (PBL) is an active, student-centered form of instruction based on John Dewey's constructivism theory. In PBL, students engage in authentic, meaningful tasks and problems that emulate real-world situations. PBL is particularly effective for achieving durable, contextual outcomes for computing students and engaging students in a sustained, collaborative focus on a specific project [46,47]. Research has also indicated that computer-assisted PBL supports numerous learning activities that engage students in a continual collaborative process of building and reshaping understanding [37,48–52].

Bouck et al. (2022) [39] detailed practical suggestions on how to integrate computational thinking and computer science into the teaching of mathematics for students with disabilities, and they described various case studies and PBL classroom activities that contribute to teaching practice. Our research is based on an in-school mathematics enrichment project for gifted children that assists students in learning mathematical concepts and computational thinking and in strengthening mathematical problem-solving skills by using the Scratch or Python programming language.

Positive behavior support (PBS) is a behavioral intervention developed by Horner and Sugai in the 1990s [53]. It is a systematic and comprehensive approach that helps students with attentional difficulties by focusing on the immediate antecedents of attentional behavior, teaching replacement behaviors, and reducing the likelihood of attentional behavior. This approach improves students' adaptive learning styles [54–56].

In recent years, PBS has also been applied to children with Autism Spectrum Disorder (ASD) and has become an effective practice for reducing challenging behaviors [57]. The intervention consists of four main components: (1) conducting a functional behavior assessment to identify the purpose and function of problem behaviors, (2) developing a behavior support plan that includes positive behavior expectations, teaching and reinforcing replacement behaviors, and modifying the environment to support positive behavior, (3) implementing the behavior support plan consistently and systematically, and (4) evaluating the effectiveness of the plan and making adjustments as needed [58]. PBS has been found to be effective in reducing problem behaviors, improving social skills and academic performance, and promoting independence and quality of life for individuals with disabilities [59].

### 1.7. Research Purposes

This study focused on providing a learning opportunity to a student with the co-morbidity of Autism Spectrum Disorder and ADHD (ASD/ADHD) who was gifted in mathematics but whose academic performance was below expectation because of 2e traits. To explore whether computer-assisted PBL instruction can help him improve concentration and academic performance, the study focused on the following three research questions:

1. Did enrollment in the computer-assisted enrichment program lead to a reduction in the frequency of inappropriate behavior in the classroom?
2. Did the 2e student improve his math problem solving and assignment completion after engaging in the program?
3. How did the 2e student's gifted peers respond to his learning behaviors and performance?

## 2. Materials and Methods

### 2.1. Background

In Taiwan, the Special Education Act provides flexibility and inclusivity in meeting the educational needs of students with special education requirements. The primary placement options depend on different educational stages, including centralized special education classes, decentralized resource rooms, mobile programs, and special education projects. Schools at all levels are encouraged to integrate relevant resources and may hire professionals to assist in teaching, aiming to fully unleash the potential of special education students. Additionally, we offer tailored gifted education programs for students with exceptional mathematical abilities, utilizing pull-out programs and other enrichment courses [11].

It is worth noting that some of these students are twice-exceptional students, particularly those who have conditions such as Autism Spectrum Disorder (ASD) or Attention Deficit Hyperactivity Disorder (ADHD). Research has found that approximately 20% of these students possess the capability to demonstrate exceptional mathematical skills alongside their conditions [60].

In line with these initiatives, two of the researchers in this study are experienced teachers who have served in the New Taipei City Gifted Education Counseling Group for eighteen and fourteen years, respectively.

This study focuses on an in-school math enrichment program for students with mathematical talent, including one student with ASD/ADHD. The program involves the use of computer programming languages, such as Scratch and Python, to enhance their computational thinking skills, understanding of mathematical concepts, and problem-solving abilities. The program consists of five sessions per semester, with each session lasting for two hours and held after school. Importantly, these students participate in regular mainstream courses for all other subjects, in addition to their math curriculum. This integration allows them to interact and learn alongside their non-gifted peers in various disciplines.

Six students who were gifted in mathematics and science participated in the project; two of these students were in the ninth grade, and four were in the eighth grade. Among the four eighth-grade students, one student had been diagnosed as having ASD/ADHD by

the Special Education Students Diagnosis and Placement Counseling Committee (DPCC) of New Taipei City.

The aforementioned project used mixed-age differentiated instruction in which ninth-grade students were required to apply the knowledge learned in the eighth grade independently to design a program or game related to mathematics. The eighth-grade students were required to learn syntax commonly used in Scratch or Python in several stages under the guidance of a teacher and in accordance with their abilities. After the completion of the section, the teachers assigned suitable tasks to each student that involved simple programming syntaxes, such as loops or conditional control, to solve mathematical problems or design mathematical games.

The tasks administered in this project were based on the concepts of PBL. Targets were proposed in accordance with how the instructor and observers judged students' abilities to apply mathematics-related content, such as the design of a program that can shuffle cards (the numbers 1–13 had to be randomly arranged and could not be repeated). The participating students were required to apply the knowledge that they had learned to solve problems and submit their results to the instructor.

*2.2. Participants*

2.2.1. 2e Participant "Kent" (Pseudonym)

According to Taiwan's Special Education Act, local authorities must establish a Special Education Students' DPCC, inviting experts, educational administrators, delegates of teacher organizations, parents, special education professionals, and delegates of related institutions and groups to participate in diagnosis, placement, replacement, and counseling. The participant who was the focus of this research was an eighth-grade student identified as having high-functioning ASD and Attention Deficit/Hyperactivity Disorder by the Special Education Students' DPCC of New Taipei City, Taiwan. In this study, we will use the pseudonym "Kent". Although he exhibited few difficulties in spoken communication, when talking with others, he focused only on his interests and ignored the ideas his peers wished to convey. For example, he interrupted the teaching of a mathematics class by repeatedly asking the instructor questions about Mars. He failed to notice the impatient facial expressions of his peers. When the instructor ignored his questions, he stood on a chair to gain the instructor's attention. According to the observations of the instructor of various subjects, in a 45 min class, he typically disrupted the class more than 10 times on average, such as by asking irrelevant questions and by beating his desk.

The enrichment program which focused on mathematical talent development was the activity in which Kent showed very high interest. It was conducted after school. The instructor and case management teacher evaluated the progress of Kent, for example, the incidence of disruptive behaviors, his level of concentration in the classroom, and the degree of experiment/task completion.

According to Item 2 of Article 6 of the Supplementary Provisions on Class Grouping and Group Study in Elementary Schools and Junior High Schools in New Taipei City, "The placement of junior high school freshmen may be arranged in an S-shape according to the order of grades, by open lottery, or by using a computerized random-number method. Students must be divided into two groups, male and female, and assigned to the class". Kent was randomly assigned to his class when he was in the seventh grade. The percentile rank of the grades of in his class was 63. Compared to his potential, he was underachieving. Because of his ASD traits, the student was excluded and bullied in the seventh grade. At that time, his homeroom teacher, case management teacher, and parents cooperated to implement appropriate interventions and counseling and tried to correct his behavior problems in the regular classroom.

2.2.2. Other Participants and Teachers

In addition to Kent, there are many relevant personnel involved in this study, which are listed below:

1.  Other Participants: Three in grade seven, and two in grade nine, each of whom is a gifted and talented student in mathematics and science identified by DPCC of New Taipei City. These students were placed in different regular classes.
2.  Instructor: The first researcher of this study, who was responsible for implementing the teaching plan of the enrichment program.
3.  Observation teacher: The second researcher was responsible for recording the situation of the program through video and handwritten records.
4.  Case management teacher: Responsible for Kent's daily tutoring, not directly involved in this enrichment program, but sometimes goes to the program site to understand Kent's learning status.
5.  Homeroom teacher: Responsible for Kent's tutoring in ordinary classes, and also Kent's English teacher.
6.  Regular math teacher: Responsible for Kent's mathematics teaching in the regular class.
7.  Other subject teachers: Responsible for Kent's teaching of regular classes.
8.  Classmates in regular class: The classmates in Kent's regular class were interviewed about Kent's interactions and behavior during all kinds of classes.

*2.3. Methods*

This research used a case study design. Case studies examine individuals' experiences within one "bounded system" and provide a thorough understanding of the particular, which is transferred when the particular is recognized across similar or diverse contexts [25,33,61–63].

This research investigated a bounded system of one cohort of students in one particular school, and the researchers analyzed data collected from interviews, educational records, and task performances.

In this study, the Flow Short Scale (FSS) was used to measure students' level of attention; this is a brief questionnaire used to assess flow experience. Flow experience is considered closely related to attention. Rheinberg et al. (2003) confirmed that FSS had a reasonable reliability with a Cronbach's alpha coefficient of 0.89 and a good test–retest reliability of 0.81 [64–67].

We use the term "misbehaviors" to define the interference or distraction of Kent during a class course, including continuously asking questions that have nothing to do with the class, constantly talking about his own ideas and interrupting the teaching process, or knocking on the table, vigorously moving the chair and class materials, and drawing or playing with his own stationery. We observed Kent's concentration through two resources; one was the frequency of misbehaviors and the other was the review result from the questionnaire of FSS.

*2.4. Research Instrument*

2.4.1. Mathematics Enrichment Project for Gifted Students

The in-school mathematics enrichment project for gifted students was designed in accordance with the Domains of Technology and Mathematics of Curriculum Guidelines of 12-Year Basic Education for Elementary Schools, Junior High, and General Senior High Schools in Taiwan [68]. These domains include the following aspects:

**Domain of Technology**

C-t-IV-4, C-t-V-1, C-t-V-2, and C-t-V-3.

Note: C = Computational Thinking, t = Technology, IV = Grade7–9, V = Grade10–12, Rightmost number = Serial Number

**Domain of Mathematics**

N-6-2, N-7-2, N-8-3, N-8-4, N-8-5, S-9-11, and N-10-6.

Note: N = Number and Quantity, S = Space and Shape, Middle number 7–9 = Grade 7–9, Rightmost number = Serial Number

2.4.2. Five Courses

Four mathematical concepts, namely, divisibility, congruence, randomization, and permutation, and four related tasks were designed to teach mathematical thinking and computational thinking when solving tasks. The mathematical problem of "Coincidence" [69] was adopted to explore the properties of integer sequences in mathematics. The material studied in the first four courses assisted students to do an investigation into Renzulli and Reis's Type III problem [70], which was the fifth course's activity.

The Type III problem is an individual or group discussion of practical issues (Individual and Small Group Investigations of Real Problems), emphasizing research on high-level problems. Its purpose is to:

1. Provide opportunities for students to apply their interests, knowledge, originality, and perseverance to a problem or research of their own choice;
2. Learn research methods and advanced knowledge;
3. Develop solutions that can make a difference;
4. Develop independent research skills such as planning, organization, resource utilization, and self-evaluation;
5. Develop perseverance, self-confidence, appreciation of creativity, and ability to communicate and express ideas [70].

The "Coincidences" cited in this study is a Number Theory problem provided by National Taiwan Normal University on the Internet. Its purpose is to guide students in thinking about the number sequence with the following phenomenon, that is, the summation of the number sequence and the string combined by the first item and the last item is exactly equal, and to explore the mathematical property behind it:

- Example (1): Arithmetic sequence $4 + 5 + \cdots + 29 = 429$;
- Example (2): Arithmetic sequence $35 + 36 + \cdots + 91 = 3591$.

This question was suitable for students to understand the difference between computational thinking and mathematical thinking. It is easy to use a program to list the answers we want to obtain. The following is written in Python syntax (Figure 1):

```python
inteval = int(input("please enter the range"))
common_difference = int(input("please enter the common difference"))
for i in range(1, inteval+1, common_difference):
    for j in range(i, inteval+1, common_difference):
        result = int((i+j)*(((j-i)/common_difference)+1)*0.5)
        if result == int(str(i)+str(j)):
            print(i, j)
```

**Figure 1.** Coincidences code in Python.

It is worth noting that the above program is not limited to an arithmetic sequence with a common difference of 1, but uses the string to be converted into a number int(str(i) + str(j)) to list all the answers under the given range and common difference. In mathematics, this method of generalizing the problem is called generalization. In current mathematical research, computer programs are often used together to enumerate several results and then discuss the underlying mathematical properties, and even change the conditions to obtain more abundant results. This is a very suitable problem for primary and secondary school students to use computational thinking to help with mathematics research. Usually, students can continue to study for a period of one year. If there are good mathematical discoveries, students are also encouraged to participate in the primary and secondary school subject exhibition.

2.4.3. Instruments to Record Concentration and Academic Performances of Kent

To investigate whether Kent reduced his disruptive behaviors while attending the activities he liked, the researchers used the average incidence of disruptive behaviors in ordinary courses as a reference, as well as recorded and compared the differences between

the performance of the student in the mathematics enrichment project. In addition to personal observation records, the observer used the Flow Short Scale (FSS) to examine the student's concentration levels.

To assess task performance and task completion, the researchers developed standards based on the curriculum of the mathematics enrichment project. Two teachers scored the works of Kent using the developed rubric to evaluate the student's work performance and work completion.

Video recording was used to investigate the interactions between the Kent and his peers. The semi-structured interviews were used to investigate the thoughts and feelings of the other students regarding the performance of Kent in the project. In addition to interviewing Kent's peers, the homeroom teacher, case management teacher, and parents of Kent were also interviewed.

### 2.4.4. Five Tasks

The content of the five tasks and related mathematical concepts comprised a sequence of integers specially designed for students to explore Type III problems [70], and Task 05 [69] was the focus of the gifted student program this semester (Table 1). The instructor expected that after instruction in the first four tasks, the participating students would have sufficient ability to analyze the problem, execute the programming, and use the programming results to explore the mathematical properties and identify the underlying theorem. Completing the five tasks helped students understand the differences between mathematical and computational thinking as well as the complementary relationship between these two concepts.

**Table 1.** Task Contents.

| Task | Title | Math Concept | Content |
|------|-------|--------------|---------|
| 01 | Factor | Divisible | Design a program that, when the user enters an integer, lists all the factors of that integer |
| 02 | Calculator | Random | Design a program that is about a computation problem with two random numbers. The user has three chances to answer the correct result |
| 03 | Shuffle | Permutation | Design a program that can randomly arrange 1–13 |
| 04 | Collatz Conjecture | Congruence | Design a program that executes Collatz Conjectur |
| 05 | Coincidences | Integer Number | Design a program to explore the problem of the summation of consecutive positive integers that are exactly to be expressed as the sum of first and last terms, call "Coincidences". Example1: 35 + 36 + 37 + . . . + 89 + 90 + 91 = 3591. Example2: 4 + 5 + 6 + . . . + 27 + 28 + 29 = 429 |

Divisibility, congruence, randomization, and permutation are common, basic, and important concepts in mathematics. Among them, divisibility and congruence are closely related. In the mathematics courses of primary and secondary schools, through the two concepts of divisibility and congruence, students can understand important mathematical knowledge such as prime numbers, composite numbers, factors, and multiples. In computer courses in primary and secondary schools, randomization and permutation are often used to assist in enumerating mathematical results within a limited range and constructing mathematical models. Through randomization and permutation, students can connect to high school permutation and probability courses. Through the design of Tasks 01–04, this enrichment program allows students to be familiar with the above four mathematical concepts so that they can use the above mathematical concepts and tools to conduct research on Task 05, which is a number sequence problem [68,71].

As shown in Table 1, Task 02 (Calculator) and Task 03 (Shuffle) allow students to experience the help of modern technology in computing by writing programs that can execute and randomly calculate two digital problems and arrange natural numbers 1 to 13 arbitrarily, showing, as a result, that computational thinking and mathematical thinking complement each other [41,42,44]. Task 1 (Factor) and Task 4 (Collatz Conjecture) allow students to understand the concepts of divisibility and congruence by writing all the factors and executing the Collatz Conjecture. It is worth noting that Collatz Conjecture is a well-known mathematical problem that has not been completely solved. This course design is not to guide students to try to prove Collatz Conjecture but to use Collatz Conjecture to familiarize students with congruence. One application of congruence is to split integers into subsets of different categories. For example, a number that is divided by 2 with a remainder of 0 to form an even number set, and a number divided by 3 with a remainder of 1, form the second calculation mentioned in the Collatz Conjecture [72,73].

### 2.4.5. Task Scores

The researchers referred to Dr. Scratch's website and other research [29,33,39,44,74,75] to develop scoring criteria in accordance with the content of this project. Six outcomes were measured in this research, namely, data representation, logical thinking, flow control, mathematical thinking, misbehavior minimization, and concentration (Table 2). Mathematical thinking was analyzed on the basis of the mathematical tasks of each class. The incidence of misbehavior was observed and compared by two project teachers. The researchers interviewed the teachers of the twice-exceptional student about the incidences of inappropriate behavior that occurred during the regular classes, such as repeatedly asking questions that were unrelated to the class, knocking on the desk, and making noise. On average, these behaviors were exhibited 10 or more times in regular classes. The incidence of inappropriate behaviors in this project was scored as follows: over 10 times (1 point), 7–9 times (1.5 points), 4–6 times (2 points), 1–3 times (2.5 points), and 0 times (3 points) (Table 2). To measure student concentration, the researchers used the FSS, in which students must self-rate their concentration [67] (Appendix A). This scale comprises 10 questions, each of which is scored on a 7-point scale; thus, the total score of the scale is 70. The scores of the student self-assessment scale (the converted concentration scores) are 1–14 (1 point), 15–28 (1.5 points), 29–42 (2 points), 43–56 (2.5 points), and 57–70 (3 points) (Table 2).

**Table 2.** Task Rubric.

| Level | Basic | | Developing | | Proficiency |
|---|---|---|---|---|---|
| Scores | 1 (point) | | 2 (points) | | 3 (points) |
| Data Representation | No variables, but several fixed values | | Operations on variables | | Operations on lists |
| Logic Thinking | If | | If else, not | | Logic operation |
| Flow Control | Sequence of blocks | | Inappropriate use of loop(repeat) | | Appropriate use of loop(repeat) |
| Mathematics Thinking | No application of the key math concept | | Inappropriate application of the key math concept | | Appropriate application of the key math concept |
| Minimize Misbehaviors | 1 | 1.5 | 2 | 2.5 | 3 |
| | Over 10 times | 7–9 times | 4–6 times | 1–3 times | 0 times |
| Concentration (Self-Assessment by the student) | 1 | 1.5 | 2 | 2.5 | 3 |
| | FSS 1–14 | FSS 15–28 | FSS 29–42 | FSS 43–56 | FSS 57–70 |

2.4.6. Determination of the Scores

The instructor and observation teachers independently graded each task item except for concentration (which was self-assessed by the students) according to the task rubric immediately after each class. The grading process was divided into three steps. In the first step, two teachers independently evaluated the dimensions of data representation, logical thinking, flow control, mathematical thinking, and misbehavior minimization from the five tasks.

The second step involved determining the scores of data representation, logical thinking, flow control, and mathematical thinking using the following rules:

1.  If two teachers grade the same item with the same score, then that score is used.
2.  If two teachers grade the same item with different scores:

    (1)    When the score difference is 1 point, the average is used as the final score.
    (2)    If the score difference exceeds 1 point, the two teachers rescore the item independently.

        (a)    If the difference after rescoring is 0 points, then that score is used.
        (b)    If the difference after scoring is 1 point, the average is used as the final score.
        (c)    If the difference still exceeds 1 point after rescoring, the researcher decides the final score.

The third step involved determining the scores of misbehavior minimization using the following rules:

1.  If two teachers grade the same item with the same score, that score is used.
2.  If two teachers grade the same item with different scores, the two teachers discuss and then determine the score.

2.4.7. Data Collection

In each class, the observation teacher assisted in video recording. Each class lasted 120 min. In the first 20 min, the teacher explained the basic programming syntaxes or mathematics concepts and then assigned a task to the participants to be completed individually within 100 min. The teacher did not provide additional assistance. During the 100 min, the researcher asked the pupils to explain how they were implementing their Scratch projects. At the end of the class, regardless of whether the task was fully completed, the programming work was uploaded to the designated to Google Cloud, and the FSS questionnaire was completed. The observation teacher observed and recorded the behaviors and performance of the participants. Sometimes, the management teacher of the twice-exceptional student attended the project to monitor the learning of the participants. This teacher did not participate in the teaching.

The first-time ratings provided by the two teachers were analyzed using Cohen's kappa and Spearman's rho. Subsequently, the final scores were determined using the criterion of "Score on Tasks" to analyze the task performance. At the end of each class, the researchers used semi-structured interviews to collect information from peers of the twice-exceptional student regarding his concentration in class, the incidence of interruptions, the level of work that was completed, and the degree of cooperation he showed. The researchers also collected information from the homeroom teacher, management teacher, and parents of the twice-exceptional student.

## 3. Results

### 3.1. Incidence of Interference

The researchers learned from the interviews with the instructors, other teachers, and classmates of the ASD/ADHD student that in a 45 min class, he often had more than ten interfering behaviors (Figure 2), with an average of one interfering behavior every 4 to 5 min. On the other hand, the instructor and observer recorded less than three

interfering behaviors in the five courses of mathematics enrichment program (Figure 3). The misbehaviors decreased significantly.

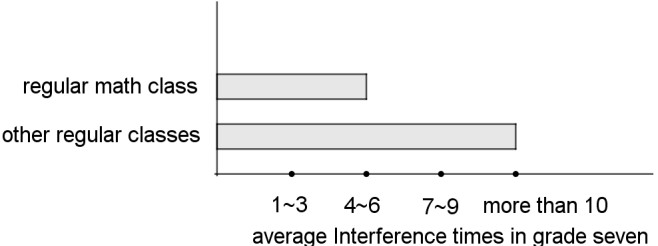

**Figure 2.** Average interference of interference in grade seven.

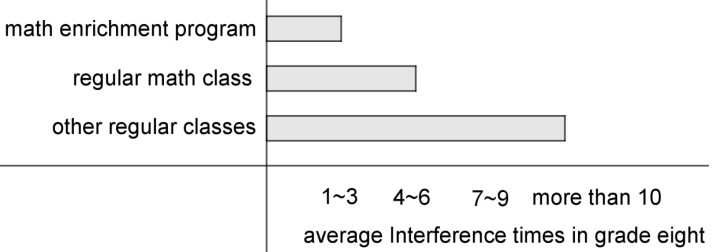

**Figure 3.** Average incidences of interference in grade eight.

Interestingly, Kent had a lower average number of interruptions in math classes and math enrichment programs than in other classes. The math teacher in the regular classes said Kent had asked irrelevant questions in math classes (Figure 3), but recently, although he did not follow the class content, he concentrated on doing the math calculations in his seat and did not interfere with the teaching. Figure 4 indicates the frequency of interruptions per month.

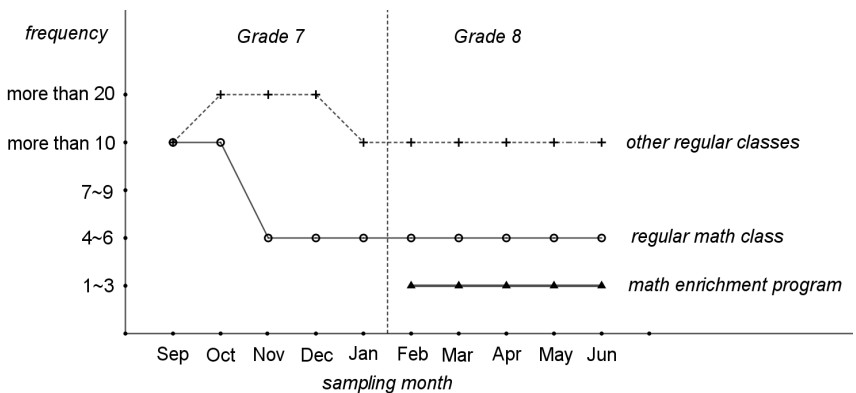

**Figure 4.** Frequency per month.

*3.2. Scores for Five Tasks*

In total, 25 items (5 tasks × 5 dimensions) were scored by the two raters during the first rating process. The inter-rater reliabilities of Cohen's kappa and Spearman's rho were 0.328 ($p = 0.001$) and 0.643 ($p = 0.001$), respectively. Table 3 indicates that all the task scores for data representation, logical thinking, mathematical thinking, misbehavior minimization, and concentration were above 2.5 points, which indicates that Kent was proficient in the basic syntax of Scratch and exhibited a sufficient level of Scratch and mathematics knowledge to complete the tasks. These results correspond with the misbehavior minimization finding that the average number of times that Kent interfered with the classroom was less than three, which was considerably lower than the typical incidence of interference during regular classes. Flow control scores indicated that Kent was still inexperienced in simplifying the

codes. Some codes were redundant, and he could have used loops to make the program more concise and readable overall. The FSS form filled out by Kent himself after completing the task indicated that he obtained full marks in the concentration dimension, which suggested that he was highly attentive when performing the tasks.

**Table 3.** Scores for the five tasks.

| No | Data Representation | Logic Thinking | Flow Control | Mathematics Thinking | Minimizing Misbehaviors | Concentration |
|---|---|---|---|---|---|---|
| 01 | 3 | 2.5 | 2.5 | 3 | 2.5 | 3 (FFS 67) |
| 02 | 3 | 3 | 2.5 | 2.5 | 2.5 | 3 (FFS 70) |
| 03 | 2.5 | 3 | 1.5 | 3 | 3 | 3 (FFS 69) |
| 04 | 3 | 3 | 1.5 | 2.5 | 2.5 | 3 (FFS 69) |
| 05 | 1 | 1.5 | 1 | 1.5 | 3 | 3 (FFS 67) |
| average | 2.5 | 2.6 | 1.8 | 2.5 | 2.7 | 3 |

Although the control process could have been more efficient, Kent completely solved Tasks 1, 2, 3, and 4. Regarding the mathematical thinking for Task 1′s factor and Task 4′s Collatz Conjecture, rather than using the mathematical concept of congruence, Kent checked whether a decimal point was present in the answer to determine "divisibility" or "nondivisibility" indirectly and successfully wrote a program that could execute factor and the Collatz Conjecture. In addressing Task 5 coincidences (Eu, 2019), Kent did not complete the program on the same day; thus, the grading teacher determined a score using the incomplete program fragments. However, the fact that Kent persevered and continued performing research after school was an encouraging sign and indicated his ambition to solve the problem.

### 3.3. Visualization of the Task Scores

Figures 5–9 illustrate the performance of Kent in the six dimensions for Tasks 1–5, respectively. Evidently, the flow control dimension (lower right of Figures 5–9) was a weak area for Kent. Moreover, this student exhibited his best performance for the concentration dimension (upper left of Figures 5–9).

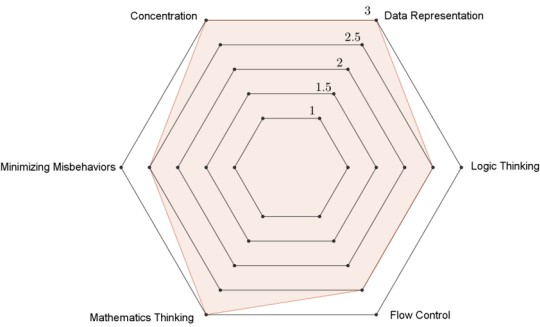

**Figure 5.** Scores for Task 1 (Factor) Basic = 1, between basic and developing = 1.5, developing = 2, between developing and proficiency = 2.5, proficiency = 3.

### 3.4. The Task Performance of Kent

According to the interview data, Kent's peers believed that the application of Scratch by Kent was more proficient than theirs. Three eighth-grade students and one ninth-grade student believed that the task completion of Kent was superior to theirs. The teacher who conducted the project reported that the task completion of Kent was higher than that of the other three students of the same grade. The teacher also noted that among the four eighth-grade students, only Kent had the energy to contribute creative patterns or animations during the task; for example, he added a flashing background and an apple picture in Task 2 (Calculator).

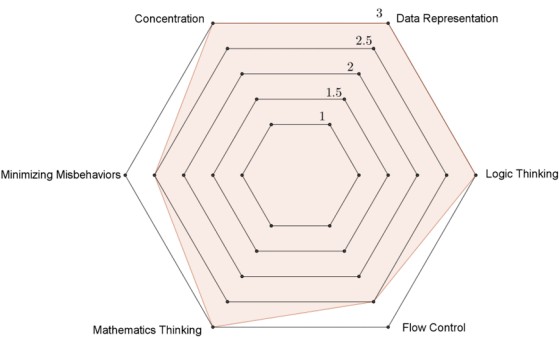

**Figure 6.** Scores for Task 2 (Calculator) Basic = 1, between basic and developing = 1.5, developing = 2, between developing and proficiency = 2.5, proficiency = 3.

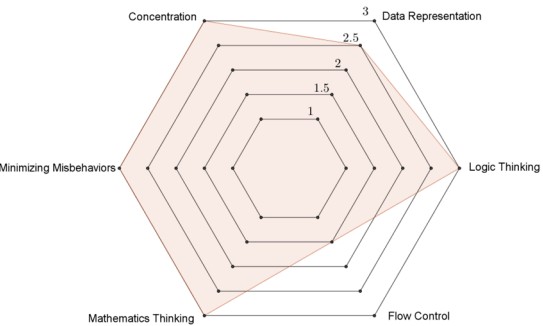

**Figure 7.** Scores for Task 3 (Shuffle). Basic = 1, between basic and developing = 1.5, developing = 2, between developing and proficiency = 2.5, proficiency = 3.

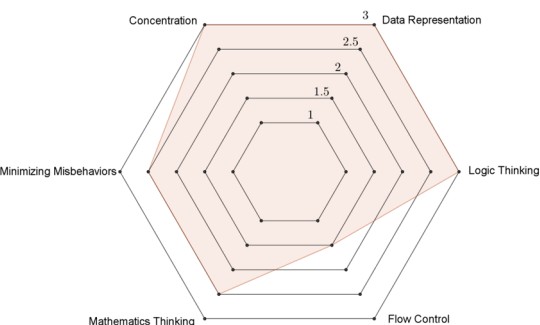

**Figure 8.** Scores for Task 4 (Collatz Conjecture) Basic = 1, between basic and developing = 1.5, developing = 2, between developing and proficiency = 2.5, proficiency = 3.

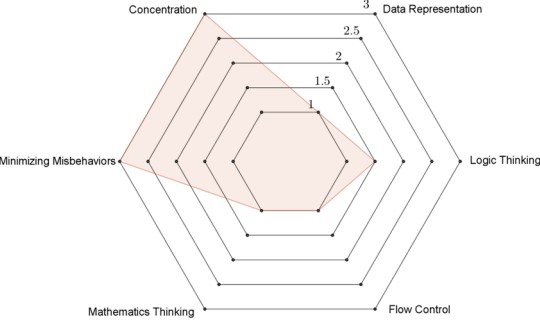

**Figure 9.** Scores for Task 5 (Coincidences) Basic = 1, between basic and developing = 1.5, developing = 2, between developing and proficiency = 2.5, proficiency = 3.

### 3.5. The Cooperation and Concentration of Kent

Regarding concentration, incidence of interference, and feelings about Kent, his peers believed that he was highly attentive in this project and exhibited no disruptive behavior.

Kent exhibited a considerably lower frequency of interruptive behaviors in the mathematics enrichment project (as observed by the project teachers) than in regular courses. The five peers responded that Kent could fluently explain the program that he wrote. For example, other students spoke as follows about Task 1:

*Student 01: "How did you do that?"*

*Kent: "It's easy. You can test the decimal point. This one (pointing to the contain block). If you exclude the one with the decimal point, the number that is not excluded is its factor."*

The case management teacher considered the incidence of disruptive behaviors by Kent in the mathematics enrichment program. Distractions were almost nonexistent in the mathematical enrichment project. The case management teacher believed that the task-based teaching method helped Kent concentrate on solving the immediate tasks and reduced his incidence of disruptive behavior. The student, his parents, and his management teacher reported that he enjoyed the project.

*Case management teacher: "This type of task-based class is really suitable for him. During other classes, he was playing with the experimental equipment and was distracted all the time. I did not expect him to be so attentive when programming."*

### 3.6. The Response from the Student's Homeroom Teacher

The homeroom teacher talked about the student's interpersonal relationships in regular classes. She mentioned that the student was bullied because of his autistic traits. The homeroom teacher and case management teacher assisted the student to learn stably in his regular class.

*The researcher: "How are his relationships in the class?"*

*Homeroom teacher: "Because of his autism, he often forgets things and keeps asking irrelevant questions that disturb the class. His interpersonal relationships are poor. He was bullied when he was in the seventh grade. Miss Yan and I immediately addressed his classmates and guided him to ask questions at the appropriate times and an appropriate number of questions, and so his relationship with his peers has improved."*

### 3.7. Details of the Task Performance of Kent
#### 3.7.1. Computational Thinking

Kent exhibited satisfactory performance in data representation, logical thinking, and mathematical thinking; however, because of a lack of experience in flow control, his codes were not streamlined, which resulted in redundancy that reduced readability (Figure 10A,B). For example, in Task 3, which involved designing a program that can randomly arrange the numbers 1–13, Kent used 13 conditional controls and the random block to generate arrangements but did not use the concept of the loop (repeat until) to simplify the codes. Thus, in the following semester, project teachers should further guide Kent in simplifying codes and combining mathematical thinking with computational thinking. To attempt all methods to solve problems, although not professional, is what teachers want to see and is in line with the constructivist viewpoint.

*Teacher: "How did you complete the shuffle program?"*

*Kent: "It is easy. You first need to specify 13 variables and then randomize them. If the first one equals the second one, randomize the second one again..."*

*Teacher: "You did a great job. Now imagine that I need to shuffle 100 cards. How will you design your program?"*

*Kent: "Well...I'll try another way."*

Although Kent did not successfully write another more efficient program to execute shuffle, he exhibited a focused expression and repeatedly attempted to solve the shuffle task. He was aware of the meaning underlying the teacher's question; the program that

he used for the arrangement of 13 numbers was too lengthy. Applying the same method to a permutation of 100 numbers would have been overly time consuming, and renewed effort was required to identify a more efficient process. The P versus NP problem is one of the fundamental mathematical problems of the contemporary era, and its importance has increased with the rise of powerful computers [76]. Awareness of efficiency is useful when teaching students about P and NP problems.

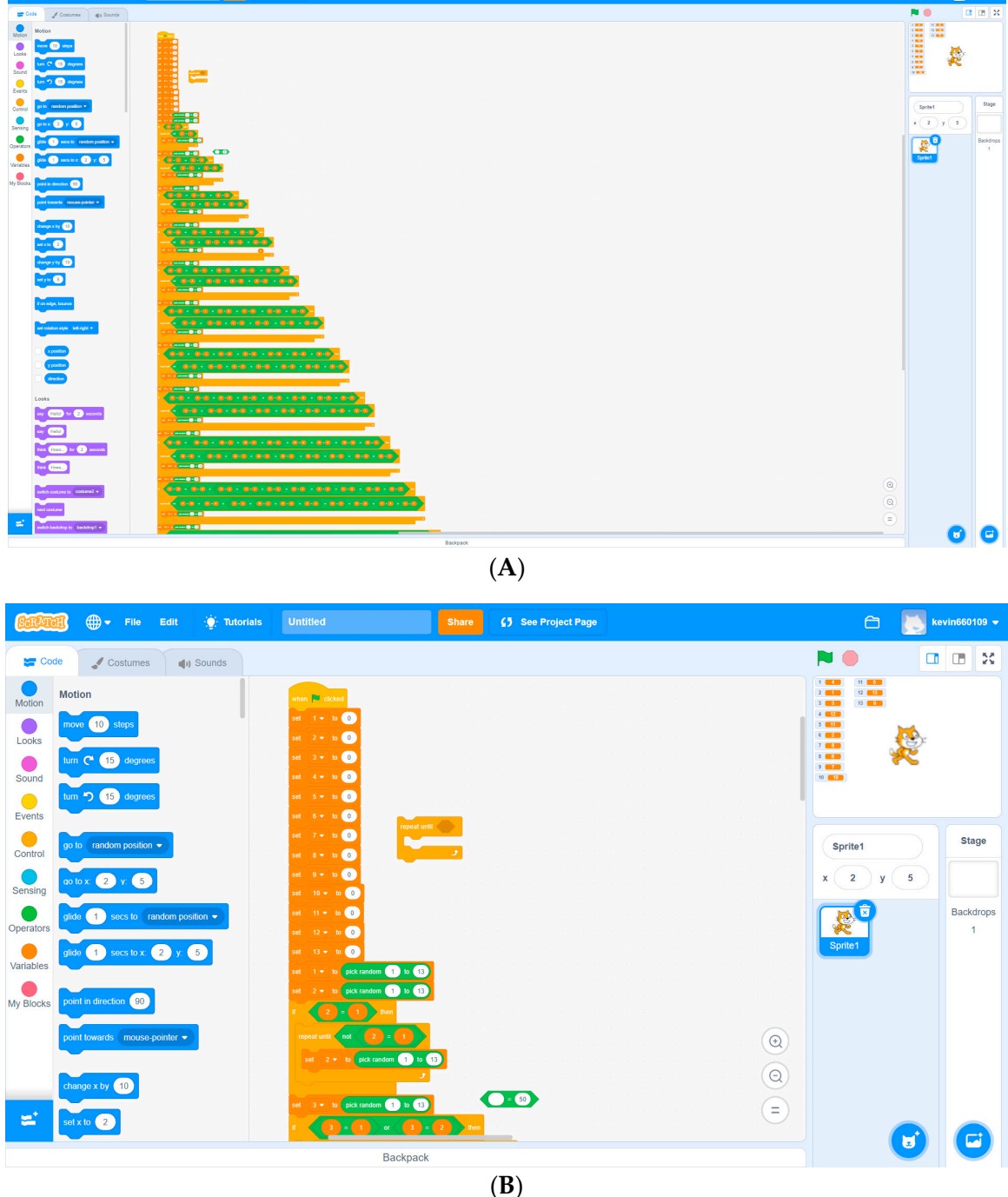

**Figure 10.** (**A**) Scratch codes generated by the twice-exceptional student in Task 3 (Shuffle). (**B**) Zoom in the scratch codes generated by the twice-exceptional student in Task 3 (Shuffle).

As depicted in Figures 11 and 12, Kent used the contain block to assess whether a number was an even or odd number to address divisibility and congruence. When the

instructor and the observer scored the mathematical thinking for the two tasks, one of the instructors believed that assessing whether a decimal point was present to replace divisibility or congruence was consistent with the mathematical thinking rubric. The observer believed that this approach was inconsistent with the underlying mathematical concept. However, the two teachers agreed that they did not expect that the twice-exceptional student would use the aforementioned method to solve tasks. In addition to exhibiting creativity, this phenomenon surprised them.

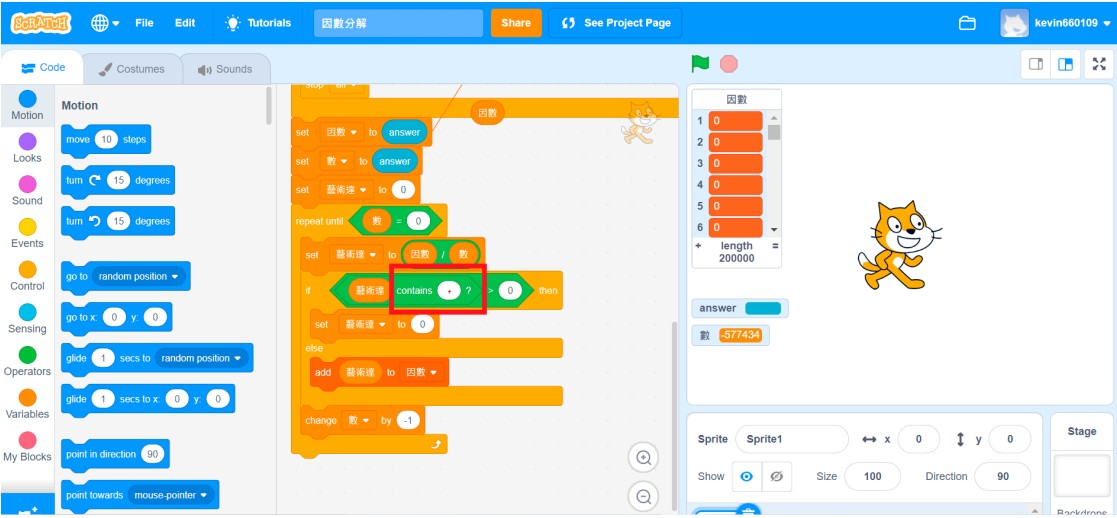

**Figure 11.** Scratch codes of factor generated by the twice-exceptional student.

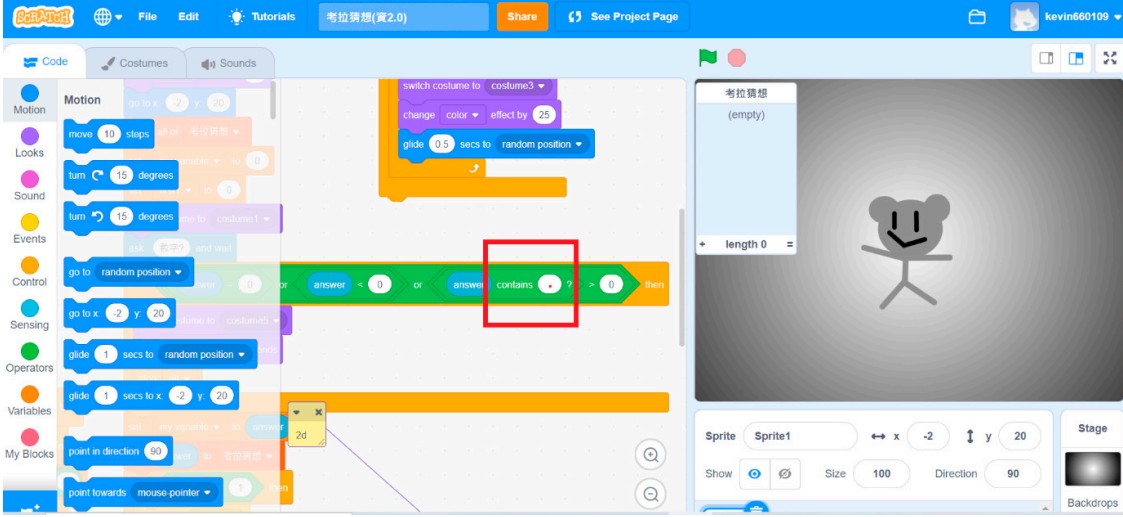

**Figure 12.** Scratch codes of the Collatz Conjecture generated by Kent.

### 3.7.2. Mathematical Thinking

In terms of randomization and permutation, as can be seen in Figure 10, Kent uses the "pick random 1 to 13" command to generate random numbers and also uses lengthy programs to achieve the result of permutation. This means that Kent is able to apply the concept of randomization and understand the meaning of permutation.

It also can be observed from Figures 13 and 14 that Kent uses various methods to deal with divisibility and congruence issues, including the "mod" command, or "greater than 0 & less than 0 & with a decimal point symbol". In the process of further interviewing Kent, it can be seen that Kent understands the concepts of divisibility and congruence, and chooses different ways to deal with the problems of divisibility and congruence according to his

own current thinking and preferences. This is also in line with the observation teacher's record; among the six project students, Kent was the only one that had time to use different methods and animation.

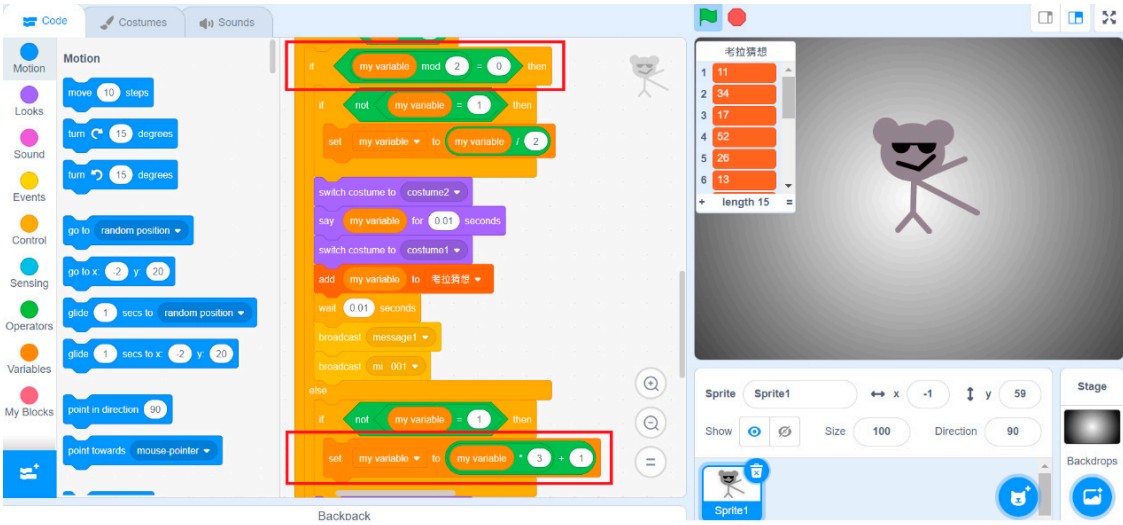

**Figure 13.** Scroll down the Scratch codes of the Collatz Conjecture generated by Kent.

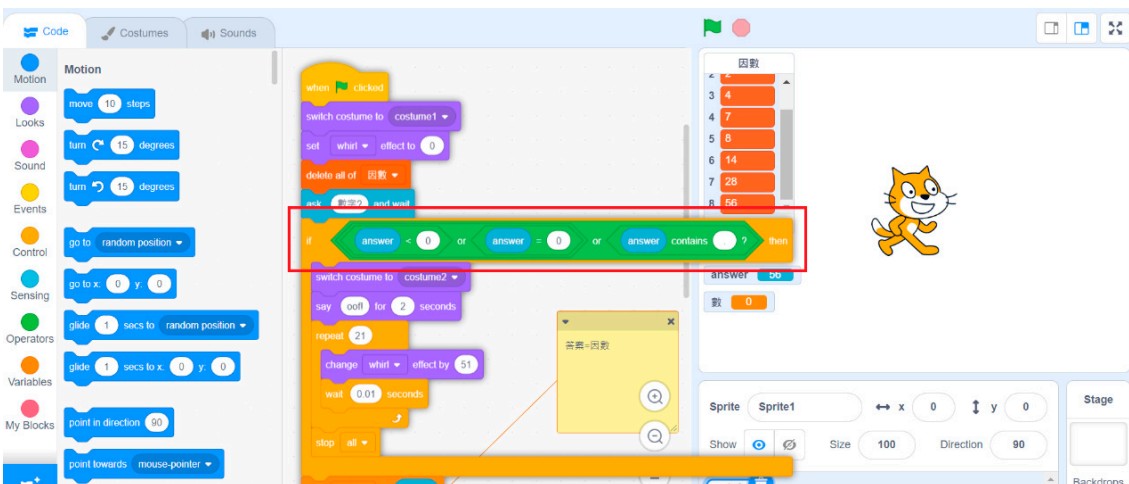

**Figure 14.** Scratch codes of the factor generated by Kent.

## 4. Discussion

In our study, the enrichment program was effective in reducing disruptive classroom behaviors for the individual case, but there is are data on the maintenance period yet as the case is still learning in the program.

The "masking effects" that giftedness and learning disabilities have on each other potentially prevent individuals from satisfying the eligibility requirements for either or both of these traits [77,78]. In this study, a "masking effect" was evident in the performance of Kent when he participated in another enrichment project in physics and chemistry. Although Kent had an aptitude for mathematics and science, because of his propensity to be easily distracted, his performance in the physics and chemistry project was considerably different from that in the mathematics enrichment project. Whether comparing his disruptive behavior or his homework completion, his performance in the mathematics enrichment project was superior to that in the physics and chemistry enrichment project.

Integrating the unique strengths, interests, and talents of each child with ASD may allow children with ASD to develop a passion for learning and the necessary skills to overcome difficulties [63,79–81]. This notion is supported by the findings of the present study.

The curriculum design of this project catered to the interests and strengths in computer programming of the twice-exceptional student. This approach reduced the incidence of interruptions by this student in the mathematics enrichment project and allowed him to demonstrate his ability to complete classroom tasks and persevere. Individuals with ASD are markedly different from one another [82]. Therefore, to apply the conclusions of this study to other classrooms, the characteristics of the students involved and their background must resemble those of this participant with ASD in this study. For example, inappropriate behavior was reduced in this study because Kent enjoys programming.

To allow students to use tools to explore and deepen their conceptual understandings [29], the project teacher in this study explained only the basic syntax, and all the tasks were completed by the students independently without further guidance. In the absence of further guidance from project teachers, all students must focus on the tasks, repeatedly attempt to complete the tasks, and reflect on their progress. Therefore, when students complete the tasks, they experience a high degree of satisfaction.

Notably, in this mathematics enrichment project during the current semester, the teacher did not provide additional explanations except for basic syntaxes; thus, all the participating students had the potential to improve their flow control. This scenario is the ideal entry point for bridging courses in the following semester to enhance the comprehensibility and simplicity of programming instructions for students.

## 5. Conclusions and Suggestion

This study aimed to help a student who is disadvantaged in multiple aspects. The case has ASD and Attention Deficit/Hyperactivity Disorder. The study investigated how computer-assisted PBL instruction minimizes misbehavior and provides the ideal entry point for deepening the understanding of mathematical concepts and computational thinking. Such a learning opportunity was very precious for the 2e student, and the learning results indicated that the content of the mathematical enrichment program was able to increase learning motivation. The after-school enrichment program, which focuses on the talent of the case, not only decreased his misbehavior in the classroom but also increased his learning achievement.

The study used interesting activities as positive support for the 2e student. Most importantly, his behavior change and achievement performances were confirmed by his teachers and peers.

In conclusion, a considerable improvement was observed in the behavior of a twice-exceptional student after PBL instruction. Incidences of misbehavior decreased from over ten times per class to less than three times per class in the mathematics enrichment project. His peers noted that he understood the mathematics concept of the tasks, was highly proficient in Scratch, completed a considerable portion of his work, and was willing to share the details of his work.

When talking about the excellence gap in education, this study is good evidence that providing appropriate learning opportunities is the best way to increase the achievement level and decrease the excellence gap.

**Author Contributions:** Conceptualization, K.-M.L., C.-C.K. and H.-L.P.; methodology, K.-M.L., C.-C.K. and H.-L.P.; formal analysis, K.-M.L.; investigation, K.-M.L. and H.-L.P.; resources, K.-M.L. and H.-L.P.; data curation, K.-M.L.; writing—original draft preparation, K.-M.L. and H.-L.P.; writing—review and editing, K.-M.L., C.-C.K. and H.-L.P.; supervision, C.-C.K.; visualization, K.-M.L. and H.-L.P.; All authors have read and agreed to the published version of the manuscript.

**Funding:** This research received no external funding.

**Institutional Review Board Statement:** Not applicable.

**Informed Consent Statement:** The respondents agreed to data use for research.

**Data Availability Statement:** The data are not publicly available due to ethical restrictions.

**Conflicts of Interest:** The authors declare no conflict of interest.

**Appendix A   [67]**

The Flow Short Scale

| Choose your option | Strongly Disagree | Disagree | Somewhat Disagree | Neither Agree Nor Disagree | Somewhat Agree | Agree | Strongly Agree |
|---|---|---|---|---|---|---|---|
| 1. I feel just the right amount of challenge | 1 | 2 | 3 | 4 | 5 | 6 | 7 |
| 2. My thoughts/ activities run fluidly and smoothly | 1 | 2 | 3 | 4 | 5 | 6 | 7 |
| 3. I do not notice time passing | 1 | 2 | 3 | 4 | 5 | 6 | 7 |
| 4. I have no difficulty concentrating | 1 | 2 | 3 | 4 | 5 | 6 | 7 |
| 5. My mind is completely clear | 1 | 2 | 3 | 4 | 5 | 6 | 7 |
| 6. I am totally absorbed in what I am doing | 1 | 2 | 3 | 4 | 5 | 6 | 7 |
| 7. The right thoughts/movements occur of their own accord | 1 | 2 | 3 | 4 | 5 | 6 | 7 |
| 8. I know what I have to do each step of the way | 1 | 2 | 3 | 4 | 5 | 6 | 7 |
| 9. I feel that I have everything under control | 1 | 2 | 3 | 4 | 5 | 6 | 7 |
| 10. I am completely lost in thought | 1 | 2 | 3 | 4 | 5 | 6 | 7 |

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
