# Peer review of "Improving Concentration and Academic Performance of a Mathematically Talented Student with ASD/ADHD: An Enrichment Program"

_education, doi:10.3390/educsci13060588_

Round 1
Reviewer 1 Report
The research seems to be up-to-date and interesting in terms of the subject. The research title is too long. But it can be shortened slightly in accordance with the purpose.
It was also good that this research was in the form of an abstracted summary. In other words, it was good to write a summary in the summary, which includes the purpose, method, data collection tool, and summary of the results. However, in the Abstract, ASD comprehension should be clearly written in parentheses.
The introduction of the study is suitable for the literature. The introduction section of the study is sufficient in terms of the subject area. The sources used are up to date. For this reason, the use of new bibliography in the introduction and discussion departments of the research enriched the research.
The aim is written in accordance with the findings of sub-objectives. The research method is well-written. Case Study Research Method was used in the study. There is sufficient information about the data collection tools used in the research, validity and reliability. 2.4.1 The title must start with capital letters.
The tables used in the research were paid to be written in the form of APA6 standard.
"Domain of Mathematics" and "Domain of Technology" topics are available in the training programs of the Ministry of National Education. There is no need to write them one by one. They can be given in a summary way in two paragraphs and must be shown in the bibliography. Only the topics in the program can be given here. Writing all is inconvenient.
"Four Mathematical Concepts, Namely Divisibility, Congruence, Randomization, and Permutation" are mentioned in the study. The research is handled in terms of technological tools. However, what is done on these issues, that is, what is done in mathematics is not included in the research. For example, I wanted to find Divisibility in Figures 11 and 12, but I couldn't find it.
As a result of the research, findings should be included in these 4 mathematics concepts. The result of the research only included training programs and technological tools outside of mathematics. These issues should be handled from the beginning and the necessary corrections should be made. In addition, Collatz's Conjecture is important. But here (EU, 2019) reference is not appropriate. In this regard, the literature should be used in the use of appropriate references and the introduction section should also be mentioned. Therefore, the tasks given in the form of figures are not understood because there were no explanations of Task 1-Task 5.
Author Response
Response to Reviewer 1 Comments
Point 1: The research seems to be up-to-date and interesting in terms of the subject. The research title is too long. But it can be shortened slightly in accordance with the purpose.
Response 1: Thank you for your suggestion. The original title was lengthy to provide the research variables. However, if we adjust the title to "Can a Strength-Based Curriculum Improve the Learning Behaviors of a Math-Talented ASD Student?" it allows for a shorter while still conveying the intended meaning. We ask your opinion on whether this revised title is clear and understandable. We highly value your input and look forward to further discussing the suitability of the new research topic.
Point 2: It was also good that this research was in the form of an abstracted summary. In other words, it was good to write a summary in the summary, which includes the purpose, method, data collection tool, and summary of the results. However, in the Abstract, ASD comprehension should be clearly written in parentheses.
Response 2: The participant with Autism Spectrum Disorder (ASD) was highly attentive during the project. (Please check line 11)
Point 3: The introduction of the study is suitable for the literature. The introduction section of the study is sufficient in terms of the subject area. The sources used are up to date. For this reason, the use of new bibliography in the introduction and discussion departments of the research enriched the research.
Response 3: Thank you so much.
Point 4: The aim is written in accordance with the findings of sub-objectives. The research method is well-written. Case Study Research Method was used in the study. There is sufficient information about the data collection tools used in the research, validity and reliability. 2.4.1 The title must start with capital letters.
Response 4: 2.4.1 Mathematics enrichment project for gifted students (Please check line 348)
Ponint 5: The tables used in the research were paid to be written in the form of APA6 standard.
Response 5: Thank you so much.
Point6: "Domain of Mathematics" and "Domain of Technology" topics are available in the training programs of the Ministry of National Education. There is no need to write them one by one. They can be given in a summary way in two paragraphs and must be shown in the bibliography. Only the topics in the program can be given here. Writing all is inconvenient.
Response 6:
Domain of Technology
C-t-IV-4, C-t-V-1, C-t-V-2, and C-t-V-3.
Note: C = Computational Thinking, t = Technology, IV = Grade7-9, V = Grade10-12, Rightmost number = Serial Number
Domain of Mathematics
N-6-2, N-7-2, N-8-3, N-8-4, N-8-5, S-9-11, and N-10-6.
Note: N = Number and Quantity, S = Space and Shape, Middle number 7-9 = Grade 7-9, Rightmost number = Serial Number
(Please check line 353-360)
Point 7: "Four Mathematical Concepts, Namely Divisibility, Congruence, Randomization, and Permutation" are mentioned in the study. The research is handled in terms of technological tools. However, what is done on these issues, that is, what is done in mathematics is not included in the research. For example, I wanted to find Divisibility in Figures 11 and 12, but I couldn't find it.
Response 7:
2.4.4 Five tasks
The content of the five tasks and related mathematical concepts comprised a sequence of integers specially designed for students to explore Type III problems [69], and Task 05 [68] was the focus of the gifted student program this semester (Table 1). The instructor expected that after instruction in the first four tasks, the participating students would have sufficient ability to analyze the problem, execute the programming, and use the programming results to explore the mathematical properties and identify the underlying theorem. Completing the five tasks helped students understand the differences between mathematical and computational thinking as well as the complementary relationship between these two concepts.
Divisibility, congruence, randomization, and permutation are common, basic, and important concepts in mathematics. Among them, divisibility and congruence are closely related. In the mathematics courses of primary and secondary schools, through the two concepts of divisibility and congruence, students can understand important mathematical knowledge such as prime numbers, composite numbers, factors, and multiples. In computer courses in primary and secondary schools, randomization, and permutation are often used to assist in enumerating mathematical results within a limited range and constructing mathematical models. Through randomization and permutation, students can connect to high school permutation and probability courses. Through the design of task 01-04, this enrichment program allows students to be familiar with the above four mathematical concepts, so that they can use the above mathematical concepts and tools to conduct research on the task 05 which is a number sequence problem [70, 71].
As shown in Table 1, Task 2 (Calculator) and Task 3 (Shuffle) allow students to experience the help of modern technology in computing by writing programs that can execute random calculate two digital problems and arrange natural numbers 1 to 13 arbitrarily, show that Computational Thinking and Mathematical Thinking complement each other as a result (Knight et al., 2019; Mohd et al., 2020; Munoz et al., 2018). Task 1 (Factor) and Task 4 (Collatz Conjecture) allow students to understand the concepts of Divisible and Congruence by writing all the factors and executing the Collatz Conjecture. It is worth noting that Collatz Conjecture is a well-known mathematical problem that has not been completely solved. This course design is not to guide students to try to prove Collatz Conjecture but to use Collatz Conjecture to familiarize students with congruence. One application of congruence is to split integers into sub-sets of different categories. For example, a number that is divided by 2 with a remainder of 0 to form an even number set, and a number divided by 3 with a remainder of 1 form the second calculation mentioned in the Collatz Conjecture [72, 73].
3.8.2 Mathematical Thinking
In terms of randomization and permutation, as can be seen in Figure 10, Kent uses the “pick random 1 to 13” command to generate random numbers and also uses lengthy programs to achieve the result of permutation. It means that Kent has been able to apply the concept of randomization and understand the meaning of permutation.
It also can be observed from Figure 13 and Figure 14 that Kent uses various methods to deal with divisibility and congruence issues, including the "mod" command, or "greater than 0 & less than 0 & with a decimal point symbol". In the process of further interviewing Kent, it can be seen that Kent understands the concepts of divisibility and congruence, and chooses different ways to deal with the problems of divisiblility and congruence according to his own current thinking and preferences. This is also in line with the observation teacher's record; among the 6 project students, Kent is the only one that has time to use different methods and animation.
Figure 13. Scroll down the Scratch codes of the Collatz Conjecture generated by Kent
Figure 14. Scratch codes of the Factor generated by Kent
His peers noted that he understood the mathematics concept of the tasks, was highly proficient in Scratch, completed a considerable portion of his work, and was willing to share the details of his work. (Please check lines 424-458, lines 692-710, and lines 764-766)
Point 8: As a result of the research, findings should be included in these 4 mathematics concepts. The result of the research only included training programs and technological tools outside of mathematics. These issues should be handled from the beginning and the necessary corrections should be made. In addition, Collatz's Conjecture is important. But here (EU, 2019) reference is not appropriate. In this regard, the literature should be used in the use of appropriate references and the introduction section should also be mentioned. Therefore, the tasks given in the form of figures are not understood because there were no explanations of Task 1-Task 5.
Response 8: Thank you for your reminder, and we added explanations in 2.4.4 and 3.8.2 and explained the mathematical results in the conclusion. (Please check lines 424-458, lines 692-710, and lines 764-766)

Reviewer 2 Report
General comment: It’s very interesting research and presentation for international readers. It can contribute to the education of students with special educational needs and, especially of those with ASD and, at the same time, ADHD. The information in the paper may help teachers and educational systems to improve classroom concentration and academic performance for their students. Authors are strongly encouraged to make corrections suggested and resubmit the paper.
Special comments:
Line 34: …Report.”: Put the full stop after the quotation mark, not before. The same in lines 40, 49. Please check throughout the text.
Line 46 Indigenous: Use lowercase i
Line 52/ Rawls and 73/Klingner: Unable to trace the authors in the references. Please check throughout the text for names that are not presented in the references list.
Line 60 In Taiwan: Use lowercase i
Lines 51-70: More references are needed to reveal the meaning of the “twice-exceptional” term. A few more lines about literature/bibliography on the phenomenon are needed. Also, the only comparison here is between Taiwan and USA. Are there a few more references about what happens in other countries? Is “twice-exceptional” an official term that is used for diagnosis (by the State/the official educational services) of such students?
Lines 73, 74: How will we spot the two authors in the final references list if there is no reference number in the in-text references? The same for “a National Education Association report (2006)” (l. 76).
Line 76: Write the country where this Association belongs.
Lines 72-94 Please strengthen bibliography in the paragraph.
Lines 207-270: The international audience (unfamiliar with Taiwan educational system) would like more information on how/if students with ASD and ADHD are coeducated with “talented/gifted in mathematics” students in the same classrooms. Is it normal for students with ASD and ADHD to be put in classrooms of advanced learners? Or “gifted in mathematics students” is a description for normal-grade students who are simply good at math and coexist with diverse-achievers in the same regular classroom? Please, provide short information, to explain if what was achieved by Kent can be achieved by (or applied to) similar special need students in regular classrooms or if it can only be applied in special needs classrooms (as implied in l. 671-674)
Line 309: …moving the chair, did not follow the class content and was drawing or playing… Write “…not following…and drawing….”
Line 311: frequency of Misbehaviors: Use lowercase -m
Line 314: start sub-title with uppercase M
Lines 320-344: Abbreviations (e.g., C-t-IV-4) are not understandable. Do they represent something?
Line 360: omit semicolon, or add it to the end of all sentences that follow (360-366).
Line 395: Concentration: Use lowercase -c
Line 413: semester(Table 1) Separate words. The same in line 427 concentration(Table 2), and lines 492, 494, 499, 609.
Lines 572-573” Write: believed that he was highly attentive in this project and that he exhibited…
Line 697: Write: …but also increased…
Lines, 782, 801, 804, 841: Please, use lowercase or uppercase uniformly in words’ first letter throughout the bibliography.
Line 849: TEACHING: Use lowercases
Line 850: Write: Lancet
Very good use of language.
Author Response
Response to Reviewer 2 Comments
Point 1: Line 34: …Report.”: Put the full stop after the quotation mark, not before. The same in lines 40, 49. Please check throughout the text.
Response 1: “Education for All Global Monitoring Report”. (Please check line 34)
Point 2: Line 46 Indigenous: Use lowercase i
Response 2: Special protection on the education for indigenous peoples, (Please check line 45)
Point 3: Line 52/ Rawls and 73/Klingner: Unable to trace the authors in the references. Please check throughout the text for names that are not presented in the references list.
Response 3: According to Rawls (1971), “social and economic inequalities of wealth and authority are only just if they result in compensating benefits for everyone, particularly the least advantaged in society” [8].. Klingner (2022) noted that a gap may exist between twice-exceptional students and their peers [18]. (Please check line 51 and line 92)
Point 4: Line 60 In Taiwan: Use lowercase i
Response 4: In Taiwan, (Please check line 62)
Ponint 5: Lines 51-70: More references are needed to reveal the meaning of the “twice-exceptional” term. A few more lines about literature/bibliography on the phenomenon are needed. Also, the only comparison here is between Taiwan and USA. Are there a few more references about what happens in other countries? Is “twice-exceptional” an official term that is used for diagnosis (by the State/the official educational services) of such students?
Response 5: 1.2 Education equity for twice-exceptional students
According to Rawls (1971), “social and economic inequalities of wealth and authority are only just if they result in compensating benefits for everyone, particularly the least advantaged in society” [8]. The term “twice-exceptional” (2E) was coined by Whitmore to describe students who have extraordinary talents or cognitive abilities but are limited in their ability to develop them because of impairments.
Students with twice exceptionality are those who have coexisting giftedness and disabilities in one or more domains that need support from both gifted and disability education [9-12]. These students may also experience discrimination or prejudice because of attributes related with their physical and mental misadjustment[13]. Individuals may be disadvantaged in numerous respects and thus may be affected by multiple equity gaps [4].
In Taiwan, for the identification of students with special education needs, local authorities should set up the Special Education Students Diagnosis and Placement Counseling Committee (briefly called DPCC), inviting scholars and experts, educational and school administrators, delegates of teacher organizations, parents, professionals of special education, and delegates of related institutions and groups to participate in diagnosis, placement, replacement, and counseling.
According to Chen, Li, Zikuda, & Kuo (2022), in Taiwan, the statistics acquired from the Ministry of Education’s Special Education Transmit Net reported that there were 376 students identified as twice exceptional at the 1-12 grade levels in 2019. In a sample of 100 gifted and talented students, 1.34% of the population was identified as twice exceptional; in a sample of 100 students with disability, 0.4% of the population was identified as twice exceptional; in a sample of 100 school age students, only 0.015% of the population was identified as twice exceptional [14, 15]. This number (0.015%) is considerably lower than the estimated 6% prevalence in the United States [16]. Without suitable support and services, these students may struggle to learn and participate in school activities, thereby leading to educational inequality [16].
In addition to the United States and Taiwan, it is worth noting that other countries, such as Spain and Ireland in Europe, also address the needs of dual exceptional (gifted and special needs) students in their legislation. Spain explicitly includes provisions for the care of gifted students and their educational needs in its laws. For instance, the "Royal Decree 696/1995" (BOE, June 2) regulates the conditions for educational attention to students with temporary or permanent special needs associated with educational history, including those arising from giftedness, mental disability, or motor or sensory impairments. Meanwhile, although Ireland's laws do not specifically mention gifted students, they emphasize the provision of education and support services that are aligned with the individual needs and abilities of all students, including those with special educational needs [17].
Therefore, ensuring that twice-exceptional students receive appropriate support and services to aid in their learning and participation in school activities is integral to educational equity. (Please check lines 50-89)
Point 6: Lines 73, 74: How will we spot the two authors in the final references list if there is no reference number in the in-text references? The same for “a National Education Association report (2006)” (l. 76).
Response 6: This number (0.015%) is considerably lower than the estimated 6% prevalence in the United States [16].According to a National Education Association report in the United States[16] (Please check lines 73,74 & 95)
Point 7: Line 76: Write the country where this Association belongs.
Response 7: This number (0.015%) is considerably lower than the estimated 6% prevalence in the United States [16]. (Please check lines 73 & 74)
Point 8: Lines 72-94 Please strengthen bibliography in the paragraph.
Response 8: In addition to the United States and Taiwan, it is worth noting that other countries, such as Spain and Ireland in Europe, also address the needs of dual exceptional (gifted and special needs) students in their legislation. Spain explicitly includes provisions for the care of gifted students and their educational needs in its laws. For instance, the "Royal Decree 696/1995" (BOE, June 2) regulates the conditions for educational attention to students with temporary or permanent special needs associated with educational history, including those arising from giftedness, mental disability, or motor or sensory impairments. Meanwhile, although Ireland's laws do not specifically mention gifted students, they emphasize the provision of education and support services that are aligned with the individual needs and abilities of all students, including those with special educational needs [17]. (Please check lines 78-86)
Point 9: Lines 207-270: The international audience (unfamiliar with Taiwan educational system) would like more information on how/if students with ASD and ADHD are coeducated with “talented/gifted in mathematics” students in the same classrooms. Is it normal for students with ASD and ADHD to be put in classrooms of advanced learners? Or “gifted in mathematics students” is a description for normal-grade students who are simply good at math and coexist with diverse-achievers in the same regular classroom? Please, provide short information, to explain if what was achieved by Kent can be achieved by (or applied to) similar special need students in regular classrooms or if it can only be applied in special needs classrooms (as implied in l. 671-674)
Response 9: 2.1 Background
In Taiwan, the Special Education Act provides flexibility and inclusivity in meeting the educational needs of students with special education requirements. The primary placement options depend on different educational stages, including centralized special education classes, decentralized resource rooms, mobile programs, and special education projects. Schools at all levels are encouraged to integrate relevant resources and may hire professionals to assist in teaching, aiming to fully unleash the potential of special education students. Additionally, we offer tailored gifted education programs for students with exceptional mathematical abilities, utilizing pull-out programs and other enrichment courses [11].
It is worth noting that some of these students are twice-exceptional students, particularly those who have conditions such as Autism Spectrum Disorder (ASD) or Attention Deficit Hyperactivity Disorder (ADHD). Research has found that approximately 20% of these students possess the capability to demonstrate exceptional mathematical skills alongside their conditions [60].
In line with these initiatives, two of the researchers in this study are experienced teachers who have served in the New Taipei City Gifted Education Counseling Group for eighteen and fourteen years, respectively.
This study focuses on an in-school math enrichment program for students with mathematical talent, including one student with ASD/ADHD. The program involves the use of computer programming languages, such as Scratch and Python, to enhance their computational thinking skills, understanding of mathematical concepts, and problem-solving abilities. The program consists of five sessions per semester, with each session lasting for two hours and held after school. Importantly, these students participate in regular mainstream courses for all other subjects, in addition to their math curriculum. This integration allows them to interact and learn alongside their non-gifted peers in various disciplines. (Please check lines 226-252)
Point 10: Line 309: …moving the chair, did not follow the class content and was drawing or playing… Write “…not following…and drawing….”
Response 10: We use the term "misbehaviors" to define the interference or distraction of Kent during a class course, including continuously asking questions that have nothing to do with the class, constantly talking about his own ideas and interrupting the teaching process, or knocking on the table, vigorously moving the chair, the class content and drawing or playing with his own stationery. (Please check lines 340-343)
Point 11: Line 311: frequency of Misbehaviors: Use lowercase -m
Response 11: We observed Kent's concentration through two resources, one was the frequency of misbehaviors and the other was the review result from the questionnaire of FSS. (Please check lines 344, 345)
Point 12: Line 314: start sub-title with uppercase M
Response 12: 2.4.1 Mathematics enrichment project for gifted students (Please check line 348)
Point 13: Lines 320-344: Abbreviations (e.g., C-t-IV-4) are not understandable. Do they represent something?
Response 13: Domain of Technology
C-t-IV-4, C-t-V-1, C-t-V-2, and C-t-V-3.
Note: C = Computational Thinking, t = Technology, IV = Grade7-9, V = Grade10-12, Rightmost number = Serial Number
Domain of Mathematics
N-6-2, N-7-2, N-8-3, N-8-4, N-8-5, S-9-11, and N-10-6.
Note: N = Number and Quantity, S = Space and Shape, Middle number 7-9 = Grade 7-9, Rightmost number = Serial Number (Please check lines 353-360)
Point 14: Line 360: omit semicolon, or add it to the end of all sentences that follow (360-366).
Response 14:
1.Provide opportunities for students to apply their interests, knowledge, originality, and perseverance to a problem or research of their own choice
2.Learn research methods and advanced knowledge
3.Develop solutions that can make a difference
4.Develop independent research skills such as planning, organization, resource utilization, and self-evaluation
5.Develop perseverance, self-confidence, appreciation of creativity, and ability to communicate and express ideas[69] (Please check lines 373- 380)
Point 15: Line 395: Concentration: Use lowercase -c
Response 15: Instruments to record concentration and academic performances of Kent (Please check line 409)
Point 16: Line 413: semester(Table 1) Separate words. The same in line 427 concentration (Table 2), and lines 492, 494, 499, 609.
Response 16: Thank you for your reminder, and we corrected it according to your suggestion. We checked all the brackets.
Point 17: Lines 572-573” Write: believed that he was highly attentive in this project and that he exhibited…
Response 17: his peers believed that he was highly attentive in this project and exhibited no disruptive behavior. (Please check lines 611,612)
Point 18: Line 697: Write: …but also increased…
Response 18: not only decreased his misbehavior in the classroom but also increased his learning achievement. (Please check lines 756,757)
Point 19: Lines, 782, 801, 804, 841: Please, use lowercase or uppercase uniformly in words’ first letter throughout the bibliography.
Response 19: Thank you for your reminder, ankd we corrected it according to your suggestion. We checked all the references.
Point 20: Line 849: TEACHING: Use lowercases
Response 20: Reis, S.M., et al., Strength-based strategies for twice-exceptional high school students with autism spectrum disorder. Teaching Exceptional Children, 2022: p. 00400599221108899. (Please check lines 930, 931)
Point 21: Line 850: Write: Lancet
Response 21: Lord, C., et al., Autism spectrum disorder. The Lancet, 2018. 392(10146): p. 508-520. (Please check line 932)

Round 2
Reviewer 1 Report
The title can be as follows.
"Improving Concentration and Academic Performance of A Mathematically Talented Student with ASD/ADHD: An Enrichment Program"
I also congratulate the authors for their corrections to make Manuscript better.